# Projecting Markov Random Field Parameters for Fast Mixing

**Xianghang Liu**
NICTA, The University of New South Wales
xianghang.liu@nicta.com.au

**Justin Domke**
NICTA, The Australian National University
justin.domke@nicta.com.au

## Abstract

Markov chain Monte Carlo (MCMC) algorithms are simple and extremely powerful techniques to sample from almost arbitrary distributions. The flaw in practice is that it can take a large and/or unknown amount of time to converge to the stationary distribution. This paper gives sufficient conditions to guarantee that univariate Gibbs sampling on Markov Random Fields (MRFs) will be fast mixing, in a precise sense. Further, an algorithm is given to project onto this set of fast-mixing parameters in the Euclidean norm. Following recent work, we give an example use of this to project in various divergence measures, comparing univariate marginals obtained by sampling after projection to common variational methods and Gibbs sampling on the original parameters.

## 1   Introduction

Exact inference in Markov Random Fields (MRFs) is generally intractable, motivating approximate algorithms. There are two main classes of approximate inference algorithms: variational methods and Markov chain Monte Carlo (MCMC) algorithms [13].

Among variational methods, mean-field approximations [9] are based on a "tractable" family of distributions, such as the fully-factorized distributions. Inference finds a distribution in the tractable set to minimize the KL-divergence from the true distribution. Other methods, such as loopy belief propagation (LBP), generalized belief propagation [14] and expectation propagation [10] use a less restricted family of target distributions, but approximate the KL-divergence. Variational methods are typically fast, and often produce high-quality approximations. However, when the variational approximations are poor, estimates can be correspondingly worse.

MCMC strategies, such as Gibbs sampling, simulate a Markov chain whose stationary distribution is the target distribution. Inference queries are then answered by the samples drawn from the Markov chain. In principle, MCMC will be arbitrarily accurate if run long enough. The principal difficulty is that the time for the Markov chain to converge to its stationary distribution, or the "mixing time", can be exponential in the number of variables.

This paper is inspired by a recent hybrid approach for Ising models [3]. This approach minimizes the divergence from the true distribution to one in a tractable family. However, the tractable family is a "fast mixing" family where Gibbs sampling is guaranteed to quickly converge to the stationary distribution. They observe that an Ising model will be fast mixing if the spectral norm of a matrix containing the absolute values of all interactions strengths is controlled. An algorithm projects onto this fast mixing parameter set in the Euclidean norm, and projected gradient descent (PGD) can minimize various divergence measures. This often leads to inference results that are better than either simple variational methods or univariate Gibbs sampling (with a limited time budget). However, this approach is limited to Ising models, and scales poorly in the size of the model, due to the difficulty of projecting onto the spectral norm.

The principal contributions of this paper are, first, a set of sufficient conditions to guarantee that univariate Gibbs sampling on an MRF will be fast-mixing (Section 4), and an algorithm to project onto this set in the Euclidean norm (Section 5). A secondary contribution of this paper is considering an alternative matrix norm (the induced $\infty$-norm) that is somewhat looser than the spectral norm, but more computationally efficient. Following previous work [3], these ideas are experimentally validated via a projected gradient descent algorithm to minimize other divergences, and looking at the accuracy of the resulting marginals. The ability to project onto a fast-mixing parameter set may also be of independent interest. For example, it might be used during maximum likelihood learning to ensure that the gradients estimated through sampling are more accurate.

## 2  Notation

We consider discrete pairwise MRFs with $n$ variables, where the i-th variable takes values in $\{1, ..., L_i\}$, $\mathcal{E}$ is the set of edges, and $\theta$ are the potentials on each edge. Each edge in $\mathcal{E}$ is an ordered pair $(i, j)$ with $i \leq j$. The parameters are a set of matrices $\theta := \{\theta^{ij} | \theta^{ij} \in \mathcal{R}^{L_i \times L_j}, \forall (i, j) \in \mathcal{E}\}$. When $i > j$, and $(j, i) \in \mathcal{E}$, we let $\theta^{ij}$ denote the transpose of $\theta^{ji}$. The corresponding distribution is

$$p(x; \theta) = \exp\left(\sum_{(i,j) \in \mathcal{E}} \theta^{ij}(x_i, x_j) - A(\theta)\right), \tag{1}$$

where $A(\theta) := \log \sum_x \exp\left(\sum_{(i,j) \in \mathcal{E}} \theta^{ij}(x_i, x_j)\right)$ is the log-partition function, and $\theta^{ij}(x_i, x_j)$ denotes the entry in the $x_i$-th row and $x_j$-th column of $\theta^{ij}$. It is easy to show that any parametrization of a pairwise MRF can be converted into this form. "Self-edges" $(i, i)$ can be included in $\mathcal{E}$ if one wishes to explicitly represent univariate terms.

It is sometimes convenient to work with the exponential family representation

$$p(x; \theta) = \exp\{f(x) \cdot \theta - A(\theta)\}, \tag{2}$$

where $f(x)$ is the sufficient statistics for configuration $x$. If these are indicator functions for all configurations of all pairs in $\mathcal{E}$, then the two representations are equivalent.

## 3  Background Theory on Rapid Mixing

This section reviews background on mixing times that will be used later in the paper.

**Definition 1.** Given two finite distributions $p$ and $q$, the **total variation distance** $\| \cdot \|_{TV}$ is defined as $\|p(X) - q(X)\|_{TV} = \frac{1}{2} \sum_x |p(X = x) - q(X = x)|$.

Next, one must define a measure of how fast a Markov chain converges to the stationary distribution. Let the state of the Markov chain after $t$ iterations be $X^t$. Given a constant $\epsilon$, this is done by finding some number of iterations $\tau(\epsilon)$ such that the induced distribution $p(X^t | X^0 = x)$ will always have a distance of less than $\epsilon$ from the stationary distribution, irrespective of the starting state $x$.

**Definition 2.** Let $\{X^t\}$ be the sequence of random variables corresponding to running Gibbs sampling on a distribution $p$. The **mixing time** $\tau(\epsilon)$ is defined as $\tau(\epsilon) = \min\{t : d(t) < \epsilon\}$, where $d(t) = \max_x \|\mathbb{P}(X^t | X^0 = x) - p(X)\|_{TV}$ is the maximum distance at time $t$ when considering all possible starting states $x$.

Now, we are interested in when Gibbs sampling on a distribution $p$ can be shown to have a fast mixing time. The central property we use is the dependency of one variable on another, defined informally as how much the conditional distribution over $X_i$ can be changed when all variables other than $X_j$ are the same.

**Definition 3.** Given a distribution $p$, the dependency matrix $R$ is defined by

$$R_{ij} = \max_{x, x' : x_{-j} = x'_{-j}} \|p(X_i | x_{-i}) - p(X_i | x'_{-i})\|_{TV}.$$

Here, the constraint $x_{-j} = x'_{-j}$ indicates that all variables in $x$ and $x'$ are identical except $x_j$. The central result on rapid mixing is given by the following Theorem, due to Dyer et al. [5], generalizing the work of Hayes [7]. Informally, it states that if $\|R\| < 1$ for *any* sub-multiplicative norm $\| \cdot \|$, then mixing will take on the order of $n \ln n$ iterations, where $n$ is the number of variables.

**Theorem 4.** *[5, Lemma 17] If $\| \cdot \|$ is any sub-multiplicative matrix norm and $\|R\| < 1$, the mixing time of univariate Gibbs sampling on a system with $n$ variables with random updates is bounded by*
$$\tau(\epsilon) \leq \frac{n}{1-\|R\|} \ln\left(\frac{\|1_n\| \|1_n^T\|}{\epsilon}\right).$$

Here, $\|1_n\|$ denotes the same matrix norm applied to a matrix of ones of size $n \times 1$, and similarly for $1_n^T$. In particular, if $\| \cdot \|$ induced by a vector p-norm, then $\|1_n\| \|1_n^T\| = n$.

Since this result is true for a variety of norms, it is natural to ask, for a given matrix $R$, which norm will give the strongest result. It can be shown that for symmetric matrices (such as the dependency matrix), the spectral norm $\| \cdot \|_2$ is always superior.

**Theorem 5.** *[5, Lemma 13] If $A$ is a symmetric matrix and $\| \cdot \|$ is any sub-multiplicative norm, then $\|A\|_2 \leq \|A\|$.*

Unfortunately, as will be discussed below, the spectral norm can be more computationally expensive than other norms. As such, we will also consider the use of the $\infty$-norm $\| \cdot \|_\infty$. This leads to additional looseness in the bound in general, but is limited in some cases. In particular if $R = rG$ where $G$ is the adjacency matrix for some regular graph with degree $d$, then for all induced p-norms, $\|R\| = rd$, since $\|R\| = \max_{x \neq 0} \|Rx\|/\|x\| = r \max_{x \neq 0} \|Gx\|/\|x\| = r\|Go\|/\|o\| = rd$, where $o$ is a vector of ones. Thus, the extra looseness from using, say, $\| \cdot \|_\infty$ instead of $\| \cdot \|_2$ will tend to be minimal when the graph is close to regular, and the dependency is close to a constant value. For irregular graphs with highly variable dependency, the looseness can be much larger.

## 4 Dependency for Markov Random Fields

In order to establish that Gibbs sampling on a given MRF will be fast mixing, it is necessary to compute (a bound on) the dependency matrix $R$, as done in the following result. The proof of this result is fairly long, and so it is postponed to the Appendix. Note that it follows from several bounds on the dependency that are tighter, but less computationally convenient.

**Theorem 6.** *The dependency matrix for a pairwise Markov random field is bounded by*

$$R_{ij}(\theta) \leq \max_{a,b} \frac{1}{2}\|\theta_{\cdot a}^{ij} - \theta_{\cdot b}^{ij}\|_\infty.$$

Here, $\theta_{\cdot a}^{ij}$ indicates the $a-$th column of $\theta^{ij}$. Note that the MRF can include univariate terms as self-edges with no impact on the dependency bound, regardless of the strength of the univariate terms. It can be seen easily that from the definition of $R$ (Definition 3), for any $i$ the entry $R_{ii}$ for self-edges $(i, i)$ should always be zero. One can, without loss of generality, set each column of $\theta^{ii}$ to be the same, meaning that $R_{ii} = 0$ in the above bound.

## 5 Euclidean Projection Operator

The Euclidean distance between two MRFs parameterized respectively by $\psi$ and $\theta$ is $\|\theta - \psi\|^2 := \sum_{(i,j)\in\mathcal{E}} \|\theta^{ij} - \psi^{ij}\|_F^2$. This section considers projecting a given vector $\psi$ onto the fast mixing set or, formally, finding a vector $\theta$ with minimum Euclidean distance to $\psi$, subject to the constraint that a norm $\| \cdot \|_*$ applied to the bound on the dependency matrix $R$ is less than some constant $c$. Euclidean projection is considered because, first, it is a straightforward measure of the closeness between two parameters and, second, it is the building block of the projected gradient descent for projection in other distance measures. To begin with, we do not specify the matrix norm $\| \cdot \|_*$, as it could be any sub-multiplicative norm (Section 3).

Thus, in principle, we would like to find $\theta$ to solve

$$\text{proj}_c(\psi) := \underset{\theta:\|R(\theta)\|_*\leq c}{\text{argmin}} \|\theta - \psi\|^2. \tag{3}$$

Unfortunately, while convex, this optimization turns out to be somewhat expensive to solve, due to a lack of smoothness Instead, we introduce a matrix $Z$, and constrain that $Z_{ij} \geq R_{ij}(\theta)$, where $R_{ij}(\theta)$ is the bound on dependency in Thm 6 (as an equality). We add an extra quadratic term

$\alpha \|Z - Y\|_F^2$ to the objective, where $Y$ is an arbitrarily given matrix and $\alpha > 0$ is trade-off between the smoothness and the closeness to original problem (3). The smoothed projection operator is

$$\text{proj}_{\mathcal{C}}(\psi, Y) := \underset{(\theta, Z) \in \mathcal{C}}{\text{argmin}} \|\theta - \psi\|^2 + \alpha \|Z - Y\|_F^2, \quad \mathcal{C} = \{(\theta, Z) : Z_{ij} \geq R_{ij}(\theta), \|Z\|_* \leq c\}. \quad (4)$$

If $\alpha = 0$, this yields a solution that is identical to that of Eq. 3. However, when $\alpha = 0$, the objective in Eq. 4 is not strongly convex as a function of $Z$, which results in a dual function which is non-smooth, meaning it must be solved with a method like subgradient descent, with a slow convergence rate. In general, of course, the optimal point of Eq. 4 is different to that of Eq. 3. However, the main usage of the Euclidean projection operator is the projection step in the projected gradient descent algorithm for divergence minimization. In these tasks the smoothed projection operator can be directly used in the place of the non-smoothed one without changing the final result. In situations when the exact Euclidean projection is required, it can be done by initializing $Y_1$ arbitrarily and repeating $(\theta_{k+1}, Y_{k+1}) \leftarrow \text{proj}_{\mathcal{C}}(\psi, Y_k)$, for $k = 1, 2, \ldots$ until convergence.

## 5.1 Dual Representation

**Theorem 7.** *Eq. 4 has the dual representation*

$$\begin{aligned} \underset{\sigma, \phi, \Delta, \Gamma}{maximize} \quad & g(\sigma, \phi, \Delta, \Gamma) \\ subject\ to \quad & \sigma_{ij}(a, b, c) \geq 0, \phi_{ij}(a, b, c) \geq 0, \quad \forall (i, j) \in \mathcal{E}, a, b, c \end{aligned}, \quad (5)$$

*where*

$$g(\sigma, \phi, \Delta, \Gamma) = \min_Z h_1(Z; \sigma, \phi, \Delta, \Gamma) + \min_\theta h_2(\theta; \sigma, \phi)$$

$$h_1(Z; \sigma, \phi, \Delta, \Gamma) = -tr(Z\Lambda^T) + I(\|Z\|_* \leq c) + \alpha \|Z - Y\|_F^2$$

$$h_2(\theta; \sigma, \phi) = \|\theta - \psi\|^2 + \frac{1}{2} \sum_{i,j \in \mathcal{E}} \sum_{a,b,c} \left( \sigma_{ij}(a, b, c) - \phi_{ij}(a, b, c) \right) (\theta_{c,a}^{ij} - \theta_{c,b}^{ij}),$$

*in which* $\Lambda_{ij} := \Delta_{ij} D_{ij} + \hat{\Gamma}_{ij} + \sum_{a,b,c} \sigma_{ij}(a, b, c) + \phi_{ij}(a, b, c)$, *where* $\hat{\Gamma}_{ij} := \begin{cases} \Gamma_{ij} & \text{if } (i, j) \in \mathcal{E} \\ -\Gamma_{ij} & \text{if } (j, i) \in \mathcal{E} \end{cases}$, *and $D$ is an indicator matrix with $D_{ij} = 0$ if $(i, j) \in \mathcal{E}$ or $(j, i) \in \mathcal{E}$, and $D_{ij} = 1$ otherwise. The dual variables $\sigma_{ij}$ and $\phi_{ij}$ are arrays of size $L_j \times L_i \times L_i$ for all pairs $(i, j) \in \mathcal{E}$ while $\Delta$ and $\Gamma$ are of size $n \times n$.*

The proof of this is in the Appendix. Here, $\text{I}(\cdot)$ is the indicator function with $\text{I}(x) = 0$ when $x$ is true and $\text{I}(x) = \infty$ otherwise.

Being a smooth optimization problem with simple bound constraints, Eq. 5 can be solved with LBFGS-B [2]. For a gradient-based method like this to be practical, it must be possible to quickly evaluate $g$ and its gradient. This is complicated by the fact that $g$ is defined in terms of the minimization of $h_1$ with respect to $Z$ and $h_2$ with respect to $\theta$. We discuss how to solve these problems now. We first consider the minimization of $h_2$. This is a quadratic function of $\theta$ and can be solved analytically via the condition that $\frac{\partial}{\partial \theta} h_2(\theta; \sigma, \phi) = 0$. The closed form solution is

$$\theta_{c,a}^{ij} = \psi_{c,a}^{ij} - \frac{1}{4} \left[ \sum_b \sigma_{ij}(a, b, c) - \sum_b \sigma_{ij}(b, a, c) - \sum_b \phi_{ij}(a, b, c) + \sum_b \phi_{ij}(b, a, c) \right]$$

$\forall (i, j) \in \mathcal{E}, 1 \leq a, c \leq m..$ The time complexity is linear in the size of $\psi$.

Minimizing $h_1$ is more involved. We assume to start that there exists an algorithm to quickly project a matrix onto the set $\{Z : \|Z\|_* \leq c\}$, i.e. to solve the optimization problem of

$$\min_{\|Z\|_* \leq c} \|Z - A\|_F^2. \quad (6)$$

Then, we observe that $\arg\min_Z h_1$ is equal to

$$\arg\min_Z -tr(Z\Lambda^T) + I(\|Z\|_* \leq c) + \alpha \|Z - Y\|_F^2 = \arg\min_{\|Z\|_* \leq c} \|Z - (Y + \frac{1}{2\alpha}\Lambda)\|_F^2.$$

For different norms $\| \cdot \|_*$, the projection algorithm will be different and can have a large impact on efficiency. We will discuss in the followings sections the choices of $\| \cdot \|_*$ and an algorithm for the $\infty$-norm.

Finally, once $h_1$ and $h_2$ have been solved, the gradient of $g$ is (by Danskin's theorem [1])

$$\frac{\partial g}{\partial \Delta_{ij}} = -D_{ij}\hat{Z}_{ij}, \qquad \frac{\partial g}{\partial \Gamma_{ij}} = \hat{Z}_{ji} - \hat{Z}_{ij},$$

$$\frac{\partial g}{\partial \sigma_{ij}(a,b,c)} = \frac{1}{2}(\hat{\theta}_{c,a}^{ij} - \hat{\theta}_{c,b}^{ij}) - \hat{Z}_{ij}, \qquad \frac{\partial g}{\partial \phi_{ij}(a,b,c)} = -\partial_{\sigma_{ij}(a,b,c)}g,$$

where $\hat{Z}$ and $\hat{\theta}$ represent the solutions to the subproblems.

## 5.2 Spectral Norm

When $\| \cdot \|_*$ is set to the spectral norm, i.e. the largest singular value of a matrix, the projection in Eq. 6 can be performed by thresholding the singular values of $A$ [3]. Theoretically, using spectral norm will give a tighter bound on $Z$ than other norms (Section 3). However, computing a full singular value decomposition can be impractically slow for a graph with a large number of variables.

## 5.3 $\infty$-norm

Here, we consider setting $\| \cdot \|_*$ to the $\infty$-norm, $\|A\|_\infty = \max_i \sum_j |A_{ij}|$, which measures the maximum $l_1$ norm of the rows of $A$. This norm has several computational advantages. Firstly, to project a matrix onto a $\infty$-norm ball $\{A : \|A_\infty\| \leq c\}$, we can simply project each row $a_i$ of the matrix onto the $l_1$-norm ball $\{a : \|a\|_1 \leq c\}$. Duchi et al. [4] provide a method linear in the number of nonzeros in $a$ and logarithmic in the length of $a$. Thus, if $Z$ is an $n \times n$, matrix, Eq. 6 for the $\infty$-norm can be solved in time $n^2$ and, for sufficiently sparse matrices, in time $n \log n$.

A second advantage of the $\infty$-norm is that (unlike the spectral norm) projection in Eq. 6 preserves the sparsity of the matrix. Thus, one can disregard the matrix $D$ and dual variables $\Delta$ when solving the optimization in Theorem 7. This means that $Z$ itself can be represented sparsely, i.e. we only need variables for those $(i,j) \in \mathcal{E}$. These simplifications significantly improve the efficiency of projection, with some tradeoff in accuracy.

# 6 Projection in Divergences

In this section, we want to find a distribution $p(x;\theta)$ in the fast mixing family closest to a target distribution $p(x;\psi)$ in some divergence $D(\psi,\theta)$. The choice of divergence depends on convenience of projection, the approximate family and the inference task. We will first present a general algorithmic framework based on projected gradient descent (Algorithm 1), and then discuss the details of several previously proposed divergences [11, 3].

## 6.1 General algorithm framework for divergence minimization

The problem of projection in divergences is formulated as

$$\min_{\theta \in \bar{\mathcal{C}}} D(\psi, \theta), \tag{7}$$

$D(\cdot, \cdot)$ is some divergence measure, and $\bar{\mathcal{C}} := \{\theta : \exists Z, s.t.(\theta, Z) \in C\}$, where $C$ is the feasible set in Eq. 4. Our general strategy for this is to use projected gradient descent to solve the optimization

$$\min_{(\theta, Z) \in \mathcal{C}} D(\psi, \theta), \tag{8}$$

using the joint operator to project onto $\mathcal{C}$ described in Section 5.

For different divergences, the only difference in projection algorithm is the evaluation of the gradient $\nabla_\theta D(\psi, \theta)$. It is clear that if $(\theta^*, Z^*)$ is the solution of Eq. 8, then $\theta^*$ is the solution of 7.

## 6.2 Divergences

---

**Algorithm 1** Projected gradient descent for divergence projection

---

    Initialize $(\theta_1, Z_1)$, $k \leftarrow 1$.
    **repeat**
        $\theta' \leftarrow \theta_k - \lambda \nabla_\theta D(\psi, \theta_k)$
        $(\theta_{k+1}, Z_{k+1}) \leftarrow \text{proj}_\mathcal{C}(\theta', Z_k)$
        $k \leftarrow k + 1$
    **until** *convergence*

---

In this section, we will discuss the different choices of divergences and corresponding projection algorithms.

### 6.2.1 KL-divergence

The KL-divergence $\text{KL}(\psi\|\theta) := \sum_x p(x;\psi) \log \frac{p(x;\psi)}{p(x;\theta)}$ is arguably the optimal divergence for marginal inference because it strives to preserve the marginals of $p(x;\theta)$ and $p(x;\psi)$. However, projection in KL-divergence is intractable here because the evaluation of the gradient $\nabla_\theta \text{KL}(\psi\|\theta)$ requires the marginals of distribution $\psi$.

### 6.2.2 Piecewise KL-divergence

One tractable surrogate of $\text{KL}(\psi\|\theta)$ is the piecewise KL-divergence [3] defined over some tractable subgraphs. Here, $D(\psi, \theta) := \max_{T \in \mathcal{T}} \text{KL}(\psi_T\|\theta_T)$, where $\mathcal{T}$ is a set of low-treewidth subgraphs. The gradient can be evaluated as $\nabla_\theta D(\psi, \theta) = \nabla_\theta \text{KL}(\psi_{T^*}\|\theta_{T^*})$ where $T^* = \arg\max_{T \in \mathcal{T}} \text{KL}(\psi_T\|\theta_T)$. For any $T$ in $\mathcal{T}$, $\text{KL}(\psi_T\|\theta_T)$ and its gradient can be evaluated by the junction-tree algorithm.

### 6.2.3 Reversed KL-divergence

The "reversed" KL-divergence $\text{KL}(\theta\|\psi)$ is minimized by mean-field methods. In general $\text{KL}(\theta\|\psi)$ is inferior to $\text{KL}(\psi\|\theta)$ for marginal inference since it tends to underestimate the support of the distribution [11]. Still, it often works well in practice. $\nabla_\theta \text{KL}(\theta\|\psi)$ can computed as $\nabla_\theta \text{KL}(\theta\|\psi) = \sum_x p(x;\theta)(\theta - \psi) \cdot f(x)\big(f(x) - \mu(\theta)\big)$, which can be approxi-

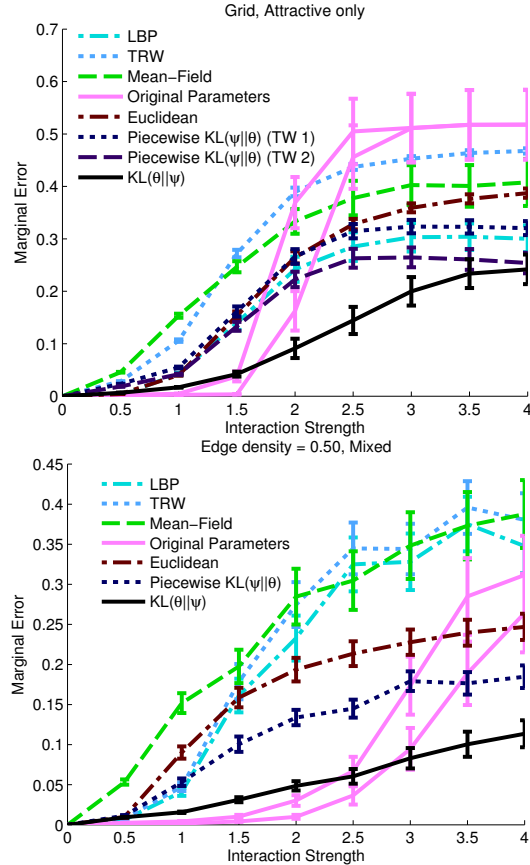

Figure 1: Mean univariate marginal error on $16 \times 16$ grids (top) with attractive interactions and median-density random graphs (bottom) with mixed interactions, comparing 30k iterations of Gibbs sampling after projection (onto the $l_\infty$ norm) to variational methods. The original parameters also show a lower curve with $10^6$ samples.

mated by samples generated from $p(x;\theta)$ [3]. In implementation, we maintain a "pool" of samples, each of which is updated by a single Gibbs step after each iteration of Algorithm 1.

## 7 Experiments

The experiments below take two stages: first, the parameters are projected (in some divergence) and then we compare the accuracy of sampling with the resulting marginals. We focus on this second aspect. However, we provide a comparison of the computation time for various projection algorithms in Table 1, and when comparing the accuracy of sampling with a given amount of time, provide two

curves for sampling with the original parameters, where one curve has an extra amount of sampling effort roughly approximating the time to perform projection in the reversed KL divergence.

## 7.1   Synthetic MRFs

Our first experiment follows that of [8, 3] in evaluating the accuracy of approximation methods in marginal inference. In the experiments, we approximate randomly generated MRF models with rapid-mixing distributions using the projection algorithms described previously. Then, the marginals of fast mixing approximate distributions are estimated by running a Gibbs chain on each distribution. These are compared against exact marginals as computed by the junction tree algorithm. We use the mean absolute difference of the marginals $|p(X_i = 1) - q(X_i = 1)|$ as the accuracy measure. We compare to Naive mean-field (MF), Gibbs sampling on original parameters (Gibbs), and Loopy belief propagation (LBP). Many other methods have been compared against a similar benchmark [6, 8].

While our methods are for general MRFs, we test on Ising potentials because this is a standard benchmark. Two graph topologies are used: two-dimensional $16 \times 16$ grids and 10 node random graphs, where each edge is independently present with probability $p_e \in \{0.3, 0.5, 0.7\}$. Node parameters $\theta_i$ are uniform from $[-d_n, d_n]$ with fixed field strength $d_n = 1.0$. Edge parameters $\theta_{ij}$ are uniform from $[-d_e, d_e]$ or $[0, d_e]$ to obtain mixed or attractive interactions respectively, with interaction strengths $d_e \in \{0, 0.5, \dots, 4\}$. Figure 1 shows the average marginal error at different interaction strengths. Error bars show the stan-

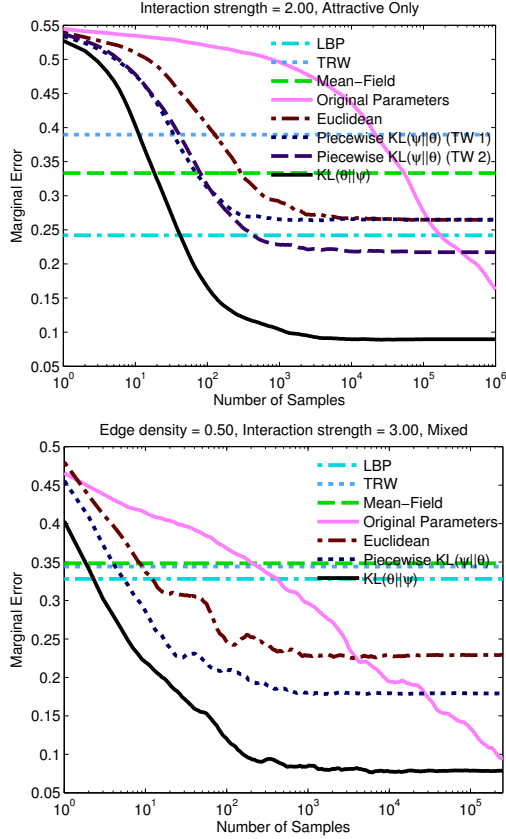

Figure 2: Examples of the accuracy of obtained marginals vs. the number of samples. Top: Grid graphs. Bottom: Median-Density Random graphs.

dard error normalized by the number of samples, which can be interpreted as a $68.27\%$ confidence interval. We also include time-accuracy comparisons in Figure 2. All results are averaged over 50 random trials. We run Gibbs long enough ($10^6$ samples) to get a fair comparison in terms of running time.

Except where otherwise stated, parameters are projected onto the ball $\{\theta : \|R(\theta)\|_\infty \le c\}$, where $c = 2.5$ is larger than the value of $c = 1$ suggested by the proofs above. Better results are obtained by using this larger constraint set, presumably because of looseness in the bound. For piecewise projection, grids use simple vertical and horizontal chains of treewidth either one or two. For random graphs, we randomly generate spanning trees until all edges are covered. Gradient descent uses a fixed step size of $\lambda = 0.1$. A Gibbs step is one "systematic-scan" pass over all variables between. The reversed KL divergence maintains a pool of 500 samples, each of which is updated by a single Gibbs step in each iteration.

We wish to compare the trade-off between computation time and accuracy represented by the choice between the use of the $\infty$ and spectral norms. We measure the running time on $16 \times 16$ grids in Table 1, and compare the accuracy in Figure 3.

The appendix contains results for a three-state Potts model on an $8 \times 8$ grid, as a test of the multivariate setting. Here, the intractable divergence $KL(\psi\|\theta)$ is included for reference, with the projection computed with the help of the junction tree algorithm for inference.

Table 1: Running times on $16 \times 16$ grids with attractive interactions. Euclidean projection converges in around 5 LBFGS-B iterations. Piecewise projection (with a treewidth of 1) and reversed KL projection use 60 gradient descent steps. All results use a single core of a Intel i7 860 processor.

| | Gibbs | | Euclidean | | Piecewise | | Reversed-KL | |
|---|---|---|---|---|---|---|---|---|
| | 30k Steps | $10^6$ Steps | $l_\infty$ norm | $l_2$ norm | $l_\infty$ norm | $l_2$ norm | $l_\infty$ norm | $l_2$ norm |
| $d_e = 1.5$ | 0.67s | 22.42s | 1.50s | 25.63s | 12.87s | 45.26s | 13.13s | 66.81s |
| $d_e = 3.0$ | 0.67s | 22.42s | 3.26s | 164.34s | 20.73s | 211.08s | 20.12s | 254.25s |

## 7.2 Berkeley binary image denoising

This experiment evaluates various methods for denoising binary images from the Berkeley segmentation dataset downscaled from $300 \times 200$ to $120 \times 80$. The images are binarized by setting $Y_i = 1$ if pixel $i$ is above the average gray scale in the image, and $Y_i = -1$. The noisy image $X$ is created by setting: $X_i = \frac{Y_i+1}{2}_i(1 - t_i^{1.25}) + \frac{1-Y_i}{2}t_i^{1.25}$, in which $t_i$ is sampled uniformly from $[0, 1]$. For inference purposes, the conditional distribution $Y$ is modeled as $P(Y|X) \propto \exp\left(\beta \sum_{ij} Y_iY_j + \frac{\alpha}{2}\sum_i(2X_i-1)Y_i\right)$, where the pairwise strength $\beta > 0$ encourages smoothness. On this attractive-only Ising potential, the Swendsen-Wang method [12] mixes rapidly, and so we use the resulting samples to estimate the ground truth. The parameters $\alpha$ and $\beta$ are heuristically chosen to be $0.5$ and $0.7$ respectively.

Figure 4 shows the decrease of average marginal error. To compare running time, Euclidean and $K(\theta\|\psi)$ projection cost approximately the same as sampling $10^5$ and $4.8 \times 10^5$ samples respectively. Gibbs sampling on the original parameter converges very slowly. Sampling the approximate distributions from our projection algorithms converge quickly in less than $10^4$ samples.

## 8 Conclusions

We derived sufficient conditions on the parameters of an MRF to ensure fast-mixing of univariate Gibbs sampling, along with an algorithm to project onto this set in the Euclidean norm. As an example use, we explored the accuracy of samples obtained by projecting parameters and then sampling, which is competitive with simple variational methods as well as traditional Gibbs sampling. Other possible applications of fast-mixing parameter sets include constraining parameters during learning.

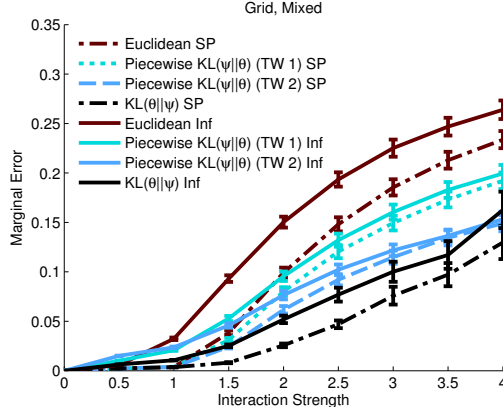

Figure 3: The marginal error using $\infty$-norm projection (solid lines) and spectral-norm projection (dotted lines) on 16x16 Ising grids.

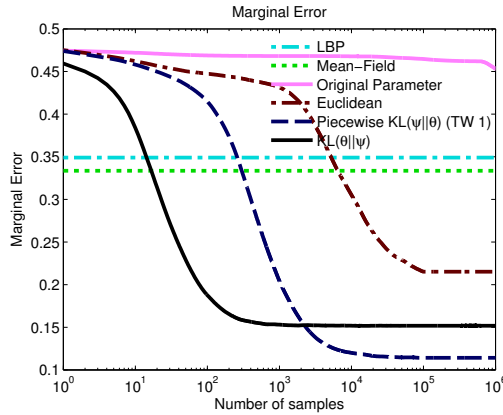

Figure 4: Average marginal error on the Berkeley segmentation dataset.

### Acknowledgments

NICTA is funded by the Australian Government through the Department of Communications and the Australian Research Council through the ICT Centre of Excellence Program.

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
