[Supplementary Material]

# 9 Appendix

## 9.1 Proof of MRF Dependency Bound

This section gives a proof of the bound on the dependency matrix stated in Section 4 above.

To start with, we observe the conditional distribution of a single variable $x_i$ when all others are fixed, which is easy to calculate.

**Lemma 8.** *The conditional probability of one variable given all others is*

$$p(X_i = \cdot | X_{-i} = x_{-i}) = sig \left( \sum_{k \in N(i)} \theta^{ik}_{\cdot x_k} \right),$$

*where sig is the "multivariate sigmoid" defined as* $(v) = \exp(v)/1^T \exp(v)$, *and* $N(i)$ *is the set of indices that are in a pair with* $i$.

Now, to compute the influence matrix, we must consider what configuration of all the variables other than $x_i$ and $x_j$ will allow a change in $x_j$ to induce the greatest change in $x_i$ (Definition 3).

**Lemma 9.** *The dependency matrix is given by*

$$R_{ij} = \max_{x,y : x_{-j} = y_{-j}} \frac{1}{2} \| sig(\theta^{ij}_{\cdot x_j} + s) - sig(\theta^{ij}_{\cdot y_j} + s) \|_1$$

$$s = \sum_{k \in N(i) \setminus j} \theta^{ik}_{\cdot x_k}$$

*Proof.* Using the previous Lemma inside the definition of the dependency matrix (Definition 3) gives that

$$R_{ij} = \max_{x,y : x_{-j} = y_{-j}} \| p(X_i = \cdot | x_{-i}) - p(X_i = \cdot | y_{-i}) \|_{TV}$$

$$= \max_{x,y : x_{-j} = y_{-j}} \frac{1}{2} \| sig( \sum_{k \in N(i)} \theta^{ik}_{\cdot x_k}) - sig( \sum_{k \in N(i)} \theta^{ik}_{\cdot y_k}) \|_1.$$

Substituting the definition of $s$ inside each of the sig() terms gives the result.

While the previous Lemma bounds the dependency, it is not in a very convenient form. Hence, the rest of this section will apply a series of relaxations to obtain more convenient upper-bounds. The first of these is obtained by letting $s$ be an arbitrary vector, rather than determined by $\theta$ and $x$. □

**Lemma 10.** *The dependency matrix for an MRF is bounded by*

$$R_{ij} \leq \max_{x_j, y_j} \max_s \frac{1}{2} \| sig(\theta^{ij}_{\cdot x_j} + s) - sig(\theta^{ij}_{\cdot y_j} + s) \|_1.$$

The following Lemma will be needed in what follows.

**Lemma 11.** *For vectors* $x, y, s$,

$$\max_s \| sig(x + s) - sig(y + s) \|_1 = 2|2a - 1|,$$

*where* $a = \sigma \left( \frac{1}{2} range(y - x) \right)$. *Here,* $range(z)$ *is defined as* $\max_i z_i - \min_i z_i$.

Now, applying this Lemma to the previous result on the dependency matrix gives the following Theorem.

**Theorem 12.** *The dependency matrix for an MRF is bounded by*

$$R_{ij} \leq \frac{1}{4} \max_{a,b} |range(\theta^{ij}_{\cdot a} - \theta^{ij}_{\cdot b})|.$$

*Proof.* The previous result gives us the bound

$$R_{ij} \le \max_{a,b} |2\sigma(\frac{1}{2}\text{range}(\theta^{ij}_{\cdot a} - \theta^{ij}_{\cdot b}) - 1|.$$

Using the easily-proven fact that $|2\sigma(\frac{1}{2}x) - 1| \le \frac{1}{4}|x|$ gives the result. □

**Corollary 13.** *The dependency matrix for an MRF is bounded by*

$$R_{ij} \le \max_{a,b} \frac{1}{4}\|\theta^{ij}_{\cdot a} - \theta^{ij}_{\cdot b}\|_1, \quad R_{ij} \le \max_{a,b} \frac{1}{2}\|\theta^{ij}_{\cdot a} - \theta^{ij}_{\cdot b}\|_\infty.$$

*Proof.* This follows immediately from the observations that $|\text{range}(x)| \le \|x\|_1$ and that $|\text{range}(x)| \le 2\|x\|_\infty$. □

## 9.2 Proof of Dual Representation for Euclidean Projection Operator

This section gives a proof of the main result of Section 5.1, as stated below.

**Theorem 14.** *The projection operator*

$$proj_{\mathcal{C}}(\psi, Y) := \underset{(\theta,Z)\in\mathcal{C}}{argmin} \|\theta - \psi\|^2 + \alpha\|Z - Y\|^2_F, \quad \mathcal{C} = \{(\theta, Z) : Z_{ij} \ge R_{ij}(\theta), \|Z\|_* \le c\} \quad (9)$$

*has the dual representation of*

$$\begin{array}{ll} \underset{\sigma,\phi,\Delta,\Gamma}{maximize} & g(\sigma, \phi, \Delta, \Gamma) \\ subject\ to & \sigma_{ij}(a, b, c) \ge 0, \phi_{ij}(a, b, c) \ge 0, \quad \forall(i, j) \in \mathcal{E}, a, b, c \end{array}, \quad (10)$$

*where*

$$g(\sigma, \phi, \Delta, \Gamma) = \min_Z h_1(Z; \sigma, \phi, \Delta, \Gamma) + \min_\theta h_2(\theta; \sigma, \phi)$$

$$h_1(Z; \sigma, \phi, \Delta, \Gamma) = -tr(Z\Lambda^T) + I(\|Z\|_* \le c) + \alpha\|Z - Y\|^2_F$$

$$h_2(\theta; \sigma, \phi) = \|\theta - \psi\|^2 + \frac{1}{2}\sum_{i,j\in\mathcal{E}}\sum_{a,b,c}\left(\sigma_{ij}(a, b, c) - \phi_{ij}(a, b, c)\right)(\theta^{ij}_{c,a} - \theta^{ij}_{c,b}),$$

*in which* $\Lambda_{ij} := \Delta_{ij}D_{ij} + \hat{\Gamma}_{ij} + \sum_{a,b,c}\sigma_{ij}(a, b, c) + \phi_{ij}(a, b, c)$, *where* $\hat{\Gamma}_{ij} :=$
$\begin{cases} \Gamma_{ij} & \text{if } (i, j) \in \mathcal{E} \\ -\Gamma_{ij} & \text{if } (j, i) \in \mathcal{E} \end{cases}$, *and D is an indicator matrix with* $D_{ij} = 0$ *if* $(i, j) \in \mathcal{E}$ *or* $(j, i) \in \mathcal{E}$,
*and* $D_{ij} = 1$ *otherwise. The dual variables* $\sigma_{ij}$ *and* $\phi_{ij}$ *are arrays of size* $L_j \times L_i \times L_i$ *for all pairs* $(i, j) \in \mathcal{E}$ *while* $\Delta$ *and* $\Gamma$ *are of size* $n \times n$.

*Proof.* Firstly, we observe that the minimization in Eq. 9 is equivalent to

$$\begin{array}{ll} \underset{\theta,Z}{minimize} & \|\theta - \psi\|^2 + \alpha\|Z - Y\|^2_F \\ subject\ to & \|Z\|_* \le c \\ & Z_{ij} = Z_{ji}, \quad \forall(i, j) \in \mathcal{E} \\ & Z_{ij} \ge \max_{1\le a,b\le m} \frac{1}{2}\|\theta^{ij}_{\cdot a} - \theta^{ij}_{\cdot b}\|_\infty, \forall(i, j) \in \mathcal{E} \\ & D_{ij}Z_{ij} = 0, \quad 1 \le i, j \le n. \end{array} \quad (11)$$

□

Now, consider the Lagrangian of this problem,

$$L(\theta, Z, \sigma, \phi, \Delta, \Gamma) := \|\theta - \psi\|^2 + \alpha\|Z - Y\|_F^2 + \mathbf{I}(\|Z\|_* \le c)$$
$$- \sum_{(i,j)\in\mathcal{E}} \sum_{a,b,c} \sigma_{ij}(a,b,c)\big(Z_{ij} - \frac{1}{2}(\theta_{c,a}^{ij} - \theta_{c,b}^{ij})\big) - \sum_{(i,j)\in\mathcal{E}} \sum_{a,b,c} \phi_{ij}(a,b,c)\big(Z_{ij} + \frac{1}{2}(\theta_{c,a}^{ij} - \theta_{c,b}^{ij})\big)$$
$$- \sum_{i,j} \Delta_{ij} D_{ij} Z_{ij} - \sum_{(i,j)\in\mathcal{E}} \Gamma_{ij}(Z_{ij} - Z_{ji}).$$

Here, $\Gamma$, $\Delta$, $\sigma_{ij}$ and $\phi_{ij}, 1 \le i, j \le n$ are dual variables and $\sum_{i,j}$ denotes $\sum_{1\le i,j\le n}$ for simplicity of notation. Here, note that $L$ is independent of $\Gamma_{ij}, \sigma_{ij}$ and $\phi_{ij}$ for $(i,j) \notin \mathcal{E}$. For convenience, one can simply set these to zero.

It is straightforward to verify that the problem in Eq. 11 is convex and Slater's conditions hold. Thus, by strong duality we have the the solution of Eq. 11 is equal to

$$\min_{\theta, Z} \max_{\sigma\ge 0, \phi\ge 0, \Delta, \Gamma} L(\theta, Z, \sigma, \phi, \Delta, \Gamma) = \max_{\sigma\ge 0, \phi\ge 0, \Delta, \Gamma} g(\sigma, \phi, \Delta, \Gamma),$$

where we define the dual function

$$g(\sigma, \phi, \Delta, \Gamma) = \min_{\theta, Z} L(\theta, Z, \sigma, \phi, \Delta, \Gamma).$$

Finally, by a simple manipulation of terms, we can see that

$$g(\sigma, \phi, \Delta, \Gamma) = \min_{Z} h_1(Z; \sigma, \phi, \Delta, \Gamma) + \min_{\theta} h_2(\theta; \sigma, \phi)$$
$$h_1(Z; \sigma, \phi, \Delta, \Gamma) = -\mathrm{tr}(Z\Lambda^T) + \mathbf{I}(\|Z\|_* \le c) + \alpha\|Z - Y\|_F^2$$
$$h_2(\theta; \sigma, \phi) = \|\theta - \psi\|^2 + \frac{1}{2} \sum_{i,j\in\mathcal{E}} \sum_{a,b,c} \big(\sigma_{ij}(a,b,c) - \phi_{ij}(a,b,c)\big)(\theta_{c,a}^{ij} - \theta_{c,b}^{ij}).$$

### 9.3 Additional Experimental Results

The rest of the appendix contains extra experimental results that could not fit in the main paper.

Figure 5: Marginal error vs. interaction strength for 3-state Potts models on grids. Here, the intractable divergence $KL(\psi\|\theta)$ is included for reference. With attractive interactions, the best-performing tractable algorithm uses the piecewise divergence, while with mixed interactions, loopy BP and simply sampling using the original parameters both perform extremely well.

Figure 6: Marginal error vs. interaction strength for Ising models on grids

Figure 7: Marginal error v.s. interaction strength for Ising models on random graphs

Figure 8: Marginal error v.s. number of samples for Ising models on grids

Figure 9: Marginal error v.s. number of samples for Ising models on random graphs with edge density 0.3

Figure 10: Marginal error v.s. number of samples for Ising models on random graphs with edge density 0.5

Figure 11: Marginal error v.s. number of samples for Ising models on random graphs with edge density 0.7