[Reviews · NeurIPS 2014]

Submitted by Assigned_Reviewer_4

This paper proposes projecting the parameters of an MRF onto the set of fast-mixing parameters: parameters for which MCMC quickly converges to the true distribution. The authors introduce a Euclidean projection operator that implements this property, but note that it can be difficult to apply. They then smooth it by requiring the projection to be close to an additional matrix input. This is sufficient for many cases, but can be applied repeatedly when the true Euclidean projection is required. The smoothed projection operator can then be applied in its dual form for faster computation. The authors also investigate the choice of norm used for the smoothing component of the projection operator.

The authors also consider non-Euclidean projections based on divergences such as the KL divergence and variations. The KL divergence itself is intractable, but surrogates can be used instead.

Experimental results show that the proposed approaches are competitive with other fast approximations for inference such as loopy belief propagation and the mean field approximation. (Do the authors also compare against tree-reweighted BP? I see "TRW" in the figures, but I do not see it defined.)

Strong points:

This is a paper that should definitely be published. It is technically rigorous and presents an innovative approach to speeding up MCMC inference in MRFs, a ubiquitous algorithm that does not always scale well.

It also opens the door for learning parameters that are fast-mixing, as the authors mention. This is an intriguing idea that could perform more accurately than learning parameters without regard to mixing and then projecting them.

Weak points:

The paper could use some proofreading. Also some notation is left undefined. Specifically:

Line 98: Distance from what? P(X^0)? I also assume total variation distance, but this could also be clarified.
Line 111: What are the maximization variables? The notation "_{-i}" is not defined.
Line 148: What are the maximization variables? The notation "_{\cdot a}" is not defined. Is this equivalent to \phi^{ij}(\cdot, a)? Is that the ath column of \phi^{ij}?

Typos:
Line 25: 'comparing of univariate' - extra 'of'
Line 97: Extra 'number'
Line 159: I think it should be \Psi instead of \phi
Line 252: 'major several'

Style file: This paper does not use the correct NIPS style file. The NIPS style is denser.

Update: Thank you for your response.
Summary: This paper proposes methods for projecting MRF parameters to the set of fast-mixing parameters. This is an innovative idea that has the potential for a significant impact.

Submitted by Assigned_Reviewer_25

The main result of the paper is a sufficient condition for fast mixing time for univariate Gibbs sampling on pairwise Markov random fields. Previous results already gave guarantees for rapid mixing if the dependency matrix is <= 1 w.r.t. any sub-multiplicative matrix norm. The condition presented in the paper are built on giving bounds on the dependency matrix using the parameters of the MRF. The condition selects a subclass of MRF parameters for which we have guaranteed rapid mixing. This gives rise to the idea of performing inference for arbitrary MRFs by first finding an MRF for which rapid mixing guarantees are satisfied and the MRF defines a distribution similar enough to the original one.
The authors investigate finding similar MRFs by projecting the parameters of MRFs using different norms and divergences.
Using different type of projections and then running Gibbs sampling on the rapidly mixing chains are evaluated on synthetic MRFs and on an image denoising task from the Berkeley segmentation data set by running the sampling algorithms for a fixed number of steps.

A few typos and comments:
Line 043: "a Markov chains"
Line 076, Eq. 1: Format properly of the outer brackets.
Line 079: Right bracket is missing from the definition of the log-partition function.
Line 098: "number number"
Line 133: I believe dividing by ||x|| is missing.
Line 166: "we would like find", "to" is missing.
Line 230: "will different" -> "will be different".
Line 232: "followings sections" -> "the following sections".

A few comments/questions to the authors:
1) Pairwise MRFs are in the focus of the paper, could similar results be stated for non-pairwise MRFs?
2) Projecting MRFs parameters using norms seem to me that might not be the right choice when trying to find an MRF describing a similar distribution, since identical distributions can be defined by different parameter vectors, furthermore, the absolute value of a parameter changes on a different scale compared to the absolute value of the corresponding marginal.
Summary: + Sufficient conditions for univariate Gibbs sampling having rapid mixing for a pairwise MRF are given.
- Only pairwise MRFs are investigated.

Submitted by Assigned_Reviewer_46

The paper considers the problem of marginal inference in discrete
MRFs. It characterizes conditions on the MRF parameters under which
a Gibbs sampler will be rapidly mixing. Further, it outlines a
procedure by which the parameters of any discrete MRF can be
projected into this family of MRFs. Some preliminary experimental
results are provided to illustrate the method.

The paper makes an important contribution to approximate marginal
inference in discrete MRFs. The ideas presented are clear and in
fact quite simple and natural in hindsight - which is a strength of
the work. The approach is unique, but draws on existing ideas,
such as approximate divergence computation using
low tree-width sub-graphs and Gibbs sampling.

There are several aspects of the work which need clarification -
the suggestions below are to help improve the presentation, make it
accessible to a wider audience, and connecting it with other
relevant related work:

i) The idea of considering a smoothed approximation to a non-smooth
problem has been studied in the literature and there are existing
results, e.g., see

Yurii Nesterov. Smooth minimization of non-smooth functions. Math.
Program., 103(1):127–152, 2005.

The characterization and discussion in Section 5 can be improved
using this literature. In particular, for suitable variants of (4),
one may be able to characterize (e.g., give bounds on) the nature
of the approximation.

ii) A brief comment on how the problem in Section 5.1 satisfies the
conditions of Danskin's theorem will be helpful to readers. Also,
please add a citation.

iii) Section 6.2.2 is interesting. Some discussion on how to choose
{\cal T} will be helpful (there is a discussion for the specific
experiments). For example, I wonder if one may be able to choose
{\cal T} to be all spanning trees and get away with tractable
computations using spanning tree LPs.

iv) The work is close in spirit to [2], so a brief discussion on
what aspects are different and/or does not directly follow from the
Ising case will be helpful.

v) Related to the above point - all experiments seem to be on Ising
models, which is covered in [2]. Some evidence of performance on
general discrete MRFs is needed.

vi) The experimental results are promising, but was hard to follow
in the first read. The proposed methodology works in two stages -
approximate the parameters, then run Gibbs sampling. A discussion
on the break-up of the run-time between these stages will be
helpful. A study of the trade-off between the number of gradient
steps in the first stage and the number of Gibbs sampling steps in
the second stage will be instructive.

vii) Over the last few years, a new approach to marginal inference
has been studied - based on solving suitable random MAP inference
problems, e.g., see the following and their references

T. Hazan and T. Jaakkola. On the partition function and random
maximum a-posteriori perturbations. ICML, 2012.

T. Hazan, S. Maji, and T. Jaakkola. On sampling from the Gibbs
distribution with random maximum a posteriori perturbations. NIPS,
2013.

A discussion and comparison with the random-MAP inference approach
will make the paper stronger.

viii) The writing is in bad shape, and seems to have been done in a
hurry. Please do a careful pass to fix the flow and presentation.
Summary: The paper presents a novel approach to marginal inference in discrete MRFs based on projection of the MRF parameters to a set where the Gibbs sampler will be rapidly mixing. The work looks promising, although work is needed to improve the presentation.
Author Feedback
Author rebuttal: We would like to thank all the reviewers for the thoughtful comments and suggestions. We look forward to revising the paper in light of this feedback.

Experiments: We definitely agree that experiments should be added with more states. The obvious generalization would be an N-state Potts model (N >= 3) with grid/random graph structure. So (absent any further feedback otherwise) we would add these results to a final version. We expect these results to be qualitatively very similar to the binary case, but this is still valuable.

Restriction to pairwise models: Absolutely, in practice, one will often want to deal with higher-order potentials. Such models can be addressed in our framework by "compiling" the higher-order model to a pairwise model (Section E.3 in Wainwright and Jordan FTML 2008) and then running our algorithm. It could be that one could gain a computational advantage by directly considering general MRFs, but we couldn't initially see one. Nevertheless, we should certainly add a discussion of this.

Projection in Norms: It is true that projection in parameter Euclidean norm is not a great choice to preserve the accuracy of the distribution. We pursue a Euclidean projection because 1) it is a relatively tractable problem (solvable as a convex optimization) and 2) it can be used as an "oracle" to project in more general divergences (Section 6) We find experimentally that using a "real" divergence essentially always outperforms a simple euclidean projection (albeit at higher computational cost).

Writing: Thanks for the many corrections. We appreciate that the writing of the paper can be improved throughout. We also were surprised to notice the issue with the style file. We are using the correct NIPS 2014 file, but in a brief comparison to other papers something is amiss, so we will correct this.

Citations: We will add and discuss the proposed citations for dual smoothing, Danskin's theorem and inference using MAP perturbations.